# The Effect of Anaerobically Cultivated Human Intestinal Microbiota Compared to Fecal Microbiota Transplantation on Gut Microbiota Profile and Symptoms of Irritable Bowel Syndrome, a Double-Blind Placebo-Controlled Study

**DOI:** 10.3390/microorganisms10091819

**Published:** 2022-09-11

**Authors:** Tarek Mazzawi, Trygve Hausken, Per Førde Refsnes, Jan Gunnar Hatlebakk, Gülen Arslan Lied

**Affiliations:** 1Division of Gastroenterology, Department of Medicine, Haukeland University Hospital, 5021 Bergen, Norway; 2Norwegian Competence Center for Functional Gastrointestinal Disorders, Section of Gastroenterology, Haukeland University Hospital, 5021 Bergen, Norway; 3Center for Nutrition, Department of Clinical Medicine, University of Bergen, 5020 Bergen, Norway; 4Faculty of Medicine, Al-Balqa Applied University, 19117 Al-Salt, Jordan

**Keywords:** ACHIM, FMT, IBS, gut microbiota, 16S rRNA sequencing

## Abstract

Fecal microbiota transplantation (FMT) from healthy donors has been shown to improve the symptoms of irritable bowel syndrome (IBS) and changes the profile of the gut microbiota for the recipients. Alternatively, anaerobically cultivated human intestinal microbiota (ACHIM) can be used to manipulate the gut microbiota. The aim of the current study was to compare the efficacy and safety of ACHIM suspension with donor-FMT and placebo (patient’s own feces) to treat IBS. Out of the 62 originally included eligible patients with diarrhea-predominant IBS and their respective donors, only 43 patients completed the study by answering the questionnaires and delivering fecal samples before transplantation and after 1, 4, 12 and 24 weeks. The patients were randomized into three subgroups for receiving ACHIM suspension (*n* = 17), donor-FMT (*n* = 11), or placebo (*n* = 15), and were followed up for 24 weeks. Fecal samples were analyzed by sequencing 16S rRNA gene using the GA-map Dysbiosis Test (Genetic Analysis AS, Oslo, Norway). IBS symptom questionnaires improved in all three subgroups. Bacterial strain signals in IBS patients were more significant for *Actinobacteria* spp. and *Bifidobacteria* spp. after receiving donor-FMT compared to placebo and for *Alistipes onderdonkii* before and after treatment in the subgroups of ACHIM and donor-FMT vs. placebo. These signals change after treatment with ACHIM suspension and donor FMT towards those measured for healthy controls, but not after placebo. IBS symptom questionnaires improved in all three forms of transplantation. Some bacterial strain signals were significantly different between ACHIM and donor-FMT vs. placebo. However, the placebo subgroup failed to change the gut microbiota towards signals measured for healthy controls. The safety and efficacy of ACHIM and donor-FMT seems similar in the current study, but further larger studies are needed.

## 1. Introduction

Irritable bowel syndrome (IBS) is a common gastrointestinal (GI) disease affecting up to 4% of the adult population [1] and leading to significant morbidity and huge costs for health care [2,3]. This is a long-lasting condition defined as abdominal pain or discomfort that is associated with a change in bowel habits [4]. Available therapies are limited but include dietary interventions, psychological support, coping strategies, and a few drug options, depending on the individual symptom characteristics (diarrhea or constipation-predominant IBS) [5,6].

The cause of IBS remains unknown, and typical pathogenetic factors hypothesized to be involved are intestinal dysmotility, visceral hypersensitivity, immunological and psychosocial factors [6]. Recent evidence suggests an important role of alterations in the gut microbiota profile [7,8] and has led to an increasing interest in probiotic and antibiotic treatment approaches [9,10]. The gut flora at the taxonomic level denominates the gut microbiota (the bacterial gene content referred to as the gut microbiome), and has been shown to be involved not only in GI diseases, but also in a variety of systemic inflammatory and metabolic diseases (e.g., atherosclerosis, obesity and type II diabetes mellitus) [11,12].

Donor fecal microbiota transplantation (donor-FMT) is infusing fecal suspension collected from a healthy donor into the GI tract of another human being to alter the gut microbiome of the new host [13,14]. It is speculated that healthy donor feces may constitute the perfect “probiotic” [15], thus proposing donor-FMT as an option for treating conditions with dysbiosis, including IBS [16,17,18]. Since 2017, donor-FMT has been generally accepted as treatment of choice for *Clostridium difficile* colitis not responding to standard antibiotic therapy [19].

Recent studies from our team [13,14] and others [20,21] investigated the effect of donor-FMT on IBS patients and have shown improvement of IBS symptoms, quality of life, gut microbiota profile [13,14,21] and duodenal enteroendocrine cells [16,17]. These studies showed that transplanted feces from healthy donors to IBS patients improved IBS symptoms for up to seven months [13,14,21].

Anaerobically cultivated human intestinal microbiota (ACHIM, ACHIM Biotherapeutics AB, Vallingby, Stockholm, Sweden) has been suggested for use as an alternative option for donor-FMT, to manipulate gut microbiota [22]. ACHIM suspension is a natural ecosystem of normal intestinal flora originating from a healthy donor in 1996. It has been inoculated anaerobically then re-cultivated every second week and stored at −70 °C, and propagated regularly in the laboratory (in vitro). Microbial analysis has repeatedly shown that ACHIM suspension still retains all major bacterial phyla; Firmicutes, Bacteroidetes, Proteobacteria and Actinobacteria [23]. Even after being cultivated for more than 20 years, ACHIM still retains its biodiversity similar to the original stool sample [23] and is free of human materials known viruses and bacterial resistance genes [24]. A previous study showed that ACHIM suspension can restore gut dysbiosis back to its ecological balance with no adverse reactions after using antibiotics in patients with recurrent *Clostridium difficile* infection causing pseudomembranous colitis [25]. Recently, it is also planned to be used to treat lower gastrointestinal symptoms in patients with systemic sclerosis [26].

In the present randomized, double-blind, placebo-controlled study, we compared the efficacy and safety of using ACHIM suspension and donor-FMT vs. placebo in relieving IBS symptoms and changing the gut microbiota profile in patients with moderate to severe IBS. 

## 2. Material and Methods

### 2.1. Eligible Patients

Patients from both genders, aged 18–65 years, who were referred between 1 January 2017 and 31 December 2018 to outpatient clinic in Haukeland University Hospital, Bergen, Norway, were included. These patients met Rome III criteria and were diagnosed with IBS with moderate to severe symptoms scoring >175 using the IBS-Symptom Severity Scale. Patients were excluded if they had a history of GI malignancy, disseminated lymphadenopathy, bloody stool, inflammatory bowel diseases (IBD), oral thrush, used antibiotics within one month before receiving FMT, were immunocompromised by taking immuno-suppressive medications, or had a history of infection with opportunistic microorganisms within one year before receiving transplantation. Pregnant or lactating women, patients who were scheduled for abdominal surgery, or patients took probiotics/antibiotics within four weeks before transplantation were also excluded from the study.

### 2.2. Randomization of Patients

The patients were randomized by a nurse who was not involved in the trial, to three subgroups according to the transplanted suspension: ACHIM, 30 g of donor-FMT or placebo (30 g of their own feces), at a ratio of 1:1:1 in blocks of six using a list generated by computer software (Sealed Envelope Ltd., London, UK). The patients delivered two fecal samples at baseline: one was used for gut microbiota analysis, while the other was either used for transplantation (randomized to placebo) or discarded (randomized to ACHIM or donor-FMT). All investigators with direct contact with participants and the patients were blinded with regard to subgroup assignment. The identity of the transplanted suspension was concealed at all times. The randomization key was revealed after the study had ended.

The study was performed in accordance with the Declaration of Helsinki [27] and was approved by the Regional Committee for Medical and Health Research Ethics in Western Norway (reference no. 2016/1914). All participants provided written informed consent. The study was registered at ClinicalTrials.gov (ID: NCT05088434) and funded by Helse Vest (project number F-10463/4800001918).

### 2.3. Donors

Healthy family members (first-grade relatives) who live within the patient’s household, from both genders, who were over 18 years old were included in a donor group. Exclusion criteria included participants who were pregnant or lactating, had a history of chronic abdominal pain, oral thrush, diarrhea, bloody stool, IBD, IBS, GI malignancy, disseminated lymphadenopathy, opportunistic infections within one year before stool donation, used antibiotic or probiotic within one month before donating stool or use immunosuppressives. 

### 2.4. Screening

All participants were scheduled for screening one week prior to transplantation day. They received study questionnaires and were physically examined then tested (with blood and stool) for inflammation, previous exposure to contagious infectious agents and other organic diseases. Serologic tests for donors included hepatitis A, B and C, Epstein-Barr virus, cytomegalovirus and human immunodeficiency virus (HIV). Patients’ serologic tests included complete blood count, creatinine with electrolytes, aspartate transaminase (AST), alanine transaminase (ALT) and international normalized ratio (INR). Stool samples from all participants were analyzed for fecal calprotectin, cultured for enteric bacterial pathogens and screened for parasites and viruses.

### 2.5. Transplantation Procedure

Gastroscopy was performed after an overnight fast to instill the transplant suspension only once followed by 60 mL of sterile 0.9% normal saline in the descending part of the duodenum distal to the papilla Vateri. All gastroscopies were performed by a gastroenterologist (T.M., G.A.L. or T.H.) at the endoscopy unit, Haukeland University Hospital, Bergen, Norway. The patients were monitored for two hours post infusion. Beyond this point, the patients reported any adverse events (bloating, abdominal pain/cramps, diarrhea, nausea, vomiting, fatigue or fever) at the time points of clinical follow-up. 

### 2.6. ACHIM Transplantation Protocol

One vial of ACHIM suspension (ACHIM Biotherapeutics AB, Vallingby, Stockholm, Sweden), which contains 30 × 10^9^ CFU of bacteria [22], was infused through the gastroscope’s work channel into the descending part of duodenum as explained above.

### 2.7. Donor-Fecal Microbiota Transplantation Protocol

Thirty grams of fresh stool samples, obtained from the donors before transplantation, was homogenized manually using 60 mL of sterile normal saline (0.9%), in accordance with previous recommendations [28,29]. Then 60 mL of the fecal suspension (screened from the patient’s sight) were infused through the gastroscope’s work channel into the descending part of the duodenum as previously mentioned.

### 2.8. Placebo Protocol

Fresh stool samples were collected from the patients and immediately infused back to them. The preparation and infusion of the fecal suspension has been previously explained under donor-FMT protocol.

### 2.9. Stool Sample Collections

The patients included in this single-center, randomized, double-blind, placebo-controlled, parallel-group study were scheduled for screening one week before the transplantation procedure and then followed up at 1, 4, 12 and 24 weeks after transplantation. Donor feces were collected at baseline to be used for transplantation for patients in the donor-FMT subgroup. The patients delivered fecal samples before the transplantation procedure then again at each scheduled visit along with completion of several study questionnaires. All of the participants were told to store the fresh stool samples in the refrigerator at 4 °C if it took more than two hours to deliver them to the endoscopy unit. All collected stool samples within 24 h were aliquoted and stored at −80 °C before they were sent for gut microbiota analysis. The participants were also informed not to change their diet or lifestyle, and to report any new incidence of infections and/or use of new medications during the study. The participants of each subgroup also completed study questionnaires at each visit before and after the transplantation.

### 2.10. Microbial DNA Analysis

Microbiota analysis was performed using the GA-map Dysbiosis Test (Genetic Analysis AS, Oslo, Norway) to obtain both gut microbiota profile and Dysbiosis Index (DI). The GA-test is based on fecal sample homogenization and mechanical bacterial cell disruption in combination with chemical cell lysis. Automated total bacterial genomic DNA extraction was performed using magnetic beads and the amplification of 16S rRNA PCR DNA covered the variable regions V3–V9. The probe was labelled by single nucleotide extension and hybridized to complementary probes coupled to magnetic beads. The detection of bacterial strain signals was performed using BioCode 1000A 128-Plex Analyzer (Applied BioCode, Santa Fe Springs, CA, USA) [30]. A Dysbiosis Index (DI) above two (maximum five) indicated that microbiota differed from the reference group (DI 1–2: non-dysbiosis, DI 3: moderate, DI 4–5: severe dysbiosis) [30]. The microbial analyses for the donors were used as healthy controls for the subgroups.

### 2.11. Study Questionnaires

Bowel habits, GI symptoms, and quality of life were recorded on the day of transplantation and at weeks 1, 4, 12 and 24 after transplantation. IBS Symptom Severity Scale (IBS-SSS) questionnaire, a 100-point visual analogue scale (VAS) that consists of five questions covering abdominal pain severity, abdominal pain frequency, abdominal distension severity, dissatisfaction with bowel habits, and interference with quality of life. The total score ranges from 0 to 500. Higher scores indicate worse symptoms; scores < 175 represent mild IBS symptoms, 175–300 represent moderate severity, >300 represent severe IBS. A reduction in IBS-SSS score by 50 points correlated with improved clinical symptoms [31]. Short form of Nepean Dyspepsia Index (SF-NDI) was used to evaluate quality of life where lower scores indicated increased quality of life [32]. Bristol Stool Form Scale (BSS) classified human feces into seven consistency categories (one for constipation and seven for diarrhea) [33,34]. Eysenck Personality Questionnaire Neuroticism (EPQ-N-12) with a cut-off value of four, scored neuroticism that is defined as a general tendency to emotional over reactivity or over responsiveness [35]. Hospital Anxiety and Depression (HAD) scale where scores greater than eight in either subscale indicated anxiety or depression [36,37,38].

## 3. Statistical Analyses

The sample size for each arm in this trial was calculated by assuming that a placebo effect was 20% and the response effects for ACHIM and donor-FMT were 80%, based on results of previous studies [21,39]. The total sample size was estimated to be 30 patients, with 10 in each arm (α = 0.05, 1 − β = 0.80). Therefore, the current study included more than 60 patients with 20 patients in each arm after we have taken into consideration the dropouts and exclusions. GraphPad Prism 7 (GraphPad Software Inc., La Jolla, CA, USA) was used for all analyses. Kruskal–Wallis with Dunn’s multiple comparison test and Paired *t*-test were used to analyze the microbial data and questionnaires, respectively, for comparison between the subgroups and for comparison between baseline before transplantation and at each visit after transplantation (*p* < 0.05 was considered to indicate a statistically significant difference).

## 4. Results

### Participants

The study originally included 62 patients with diarrhea-predominant IBS and their respective donors after fulfilling the inclusion criteria (Figure 1). None of the participants received antibiotics during the last six months prior to their inclusion to the study. The patients were randomized into three subgroups for receiving ACHIM suspension (*n* = 21), donor-FMT (*n* = 20), or placebo (*n* = 21). Nineteen of the originally recruited patients were excluded: 13 after withdrawing their consent, one underwent cholecystectomy, one became pregnant, and four used antibiotics as shown in Figure 1. Hence, 43 patients (11 males and 32 females, mean age of 36 and age range of 21–61 years) that belonged to the following subgroups: ACHIM suspension (*n* = 17), donor-FMT (*n* = 11) and placebo (*n* = 15), completed the study by answering the questionnaires and delivering fecal samples as previously explained, and were followed up for 24 weeks with regards to efficacy and safety parameters, as well as analysis of fecal samples. Blood test results were normal before the transplantation procedure and stool cultures showed no growth of any pathological organisms for either donors or patients. Blood tests for the patients were followed up again at the end of the study and did not show any pathological results. Detailed health and symptom questionnaires were provided before and after transplantation and also throughout the study. We asked both the patients and donors to report any changes in their diet, lifestyle, medications or health history during the whole study. None of the subgroups reported any such changes during the study.

## 5. Gut Microbiota

At baseline, the Dysbiosis Index for the patients in the three subgroups were normal (Figure 2). The difference in DI for patients and donors was not statistically significant. No significant changes were noted in DI for patients following transplantation in comparison with before transplantation with ACHIM, donor-FMT or placebo.

The bacterial strain signals for *Actinobacteria* spp., *Bifidobacteria* spp. and *Alistipes onderdonkii* were compared among the three subgroups and significant differences were found in the signals of *Actinobacteria* spp. between donor-FMT and placebo subgroups at visits 3 and 4 (*p* = 0.002 and 0.002, respectively) (Figure 3). Bacterial strain signals for *Bifidobacteria* spp. were significantly different between donor-FMT and placebo subgroups at visits 3 and 4 (*p* = 0.0035 and 0.0044, respectively) (Figure 3). Bacterial strain signals for *Alistipes onderdonkii* were significantly different between ACHIM and placebo subgroups at transplantation day and visits 1–4 (*p* = 0.007, 0.003, 0.002, 0.005 and 0.006, respectively) and between donor-FMT and placebo subgroups at transplantation day and visit 1 (*p* = 0.04 and 0.01, respectively) (Figure 3). 

A subanalysis of the three bacterial strain signals (*Actinobacteria* spp., *Bifidobacteria* spp. and *Alistipes onderdonkii*) for the patients before and after transplantation compared to healthy controls showed significant differences in patients belonging to the placebo subgroup (Table 1). These signals did not change towards the signals measured for healthy controls after receiving placebo transplantation. However, the bacterial strain signals changed in patients before and after receiving ACHIM or donor-FMT transplantation towards bacterial strain signals measured for healthy controls, but no significant differences were found when comparing them to the healthy controls (Table 1).

## 6. Study Questionnaires

The scores of the study questionnaires for each subgroup after receiving ACHIM, donor-FMT, or placebo are presented in Table 2, Table 3 and Table 4, respectively. The scores for IBS-SSS (Figure 4) for the patients in all three subgroups at transplantation day were considered severe (IBS-SSS score > 300). Patients with a reduction in IBS-SSS score > 50 points after transplantation are considered as responders and patients with a reduction < 50 points in IBS-SSS scores are considered as non-responders.

Significant changes were found between ACHIM vs. placebo subgroups at baseline before transplantation and visits 1–4 for HAD total scores (*p* = 0.05, 0.004, 0.02, 0.0009 and 0.01), HAD anxiety (*p* = 0.2, 0.03, 0.2, 0.02 and 0.08), HAD depression (*p* = 0.06, 0.04, 0.04, 0.01 and 0.04) and EPQ-N-12 (*p* = 0.02, 0.3, 0.08, 0.02, 0.02), respectively. Only EPQ-N-12 score between FMT vs. placebo subgroups at visit 4 showed a significant difference (*p* = 0.02). The rest of the questionnaires showed no significant differences when comparing the different subgroups with each other before transplantation and at each visit after transplantation. 

A subanalysis that compared the questionnaire scores at baseline before transplantation to scores at each visit after transplantation showed that all three subgroups had significant improvements at several visits for the following questionnaires: IBS-SSS, Bristol stool form, SF-NDI and EPQ-N-12 (Table 2, Table 3 and Table 4). However, no significant changes were found for HAD total, HAD anxiety or HAD depression scores, except for HAD anxiety score at the second visit for the subgroup that received placebo transplantation (Table 4).

## 7. Post-Transplantation Complications

No complications were reported in any of the three study arms during the study post transplantation.

## 8. Discussion

To our knowledge, this is the first study where ACHIM suspension and donor-FMT were compared in IBS patients. This double-blind placebo-controlled study included three different forms of microbiota transplantation (ACHIM suspension, donor-FMT and the patients’ own feces as placebo) to explore their safety and efficacy to relieve IBS symptoms and alter gut microbiota profile and dysbiosis. The patients of all three subgroups complained of severe IBS symptoms and low quality of life before transplantation that improved following transplantation with parallel changes in gut microbiota profiles (except for placebo).

The average of DI at the beginning and during the study for all three subgroups remained around two, which is considered normobiosis. We used the per-oral route of installation in this study to cover that upper part of the gastrointestinal tract in addition to the lower part. This route of administration has been safe in all studies [14,21], and did not worsen the DI.

In the current study, the changes in the bacterial strains following receiving donor-FMT are consistent with previous reports [13,14]. Generally, *Bifidobacteria* spp. count is either decreased [40,41] or increased [42] in IBS patients. Consistent with our previous results [13,14], the bacterial strain signals for *Actinobacteria* spp. and *Bifidobacteria* spp. in this study were also lower in IBS patients than healthy controls, which increased after receiving ACHIM and donor-FMT, but not after placebo. *Bifidobacteria* spp. are important for the mucosal barrier of the gut to prevent the crossover of pathogens [40]. Low levels of *Alistipes* spp. have been recently suggested to play an important role in the decreased responsiveness to FMT [43]. These signals changed towards those measured for healthy controls in the subgroups that received transplantation with ACHIM suspension and donor-FMT, thus making the profiles of the gut microbiota for the patients after transplantation similar to the healthy controls, which is consistent with previous results [13,14]. However, in the placebo subgroup, significant changes were noted in bacterial strain signals between patients and healthy control that failed to change towards signals measured for healthy controls, in addition to the significant differences in bacterial strain signals between placebo vs. ACHIM and donor-FMT. These findings highlight the significant differences between the gut microbiota of the subgroups that received ACHIM or donor-FMT and those of the placebo subgroup. *Bifidobacteria* spp. and *Alistipes* spp. are anaerobic bacteria and both are important for maintaining the wellbeing of the gut microenvironment, and influencing the brain-gut axis [44,45]. These bacteria increase after ingesting probiotics [44,45] and since ACHIM and donor-FMT are optimal forms of probiotics [15], this explains why these bacteria increased towards values measured for healthy controls contrary to the patient’s own feces as placebo. 

The symptoms of IBS (IBS-SSS questionnaire), stool consistency (Bristol stool form questionnaire), quality of life (SF-NDI questionnaire) and neuroticism (EPQ-N-12 questionnaire) improved during the study in all three subgroups, which is consistent with our previous publications [13,14]. Despite these positive results, there were no significant differences noted in the scores of these questionnaires when comparing among the three subgroups, meaning that the effect of the active treatments was similar to placebo, which is consistent with a previous publication by Halkjaer et al. [46].

The patients in ACHIM and donor-FMT subgroups showed significant differences in total HAD, HAD anxiety and HAD depression scores when compared to the placebo subgroup, however, this might be due to the fact that the scores of these questionnaires were low at baseline and continued this way after receiving placebo. In addition, the scores of these questionnaires showed no statistically significant difference after receiving the different transplantation suspensions compared to before transplantation, which may be a type II error due to the small sample sizes especially in the donor-FMT subgroup. The effect of FMT on psychiatric symptoms in IBS patients has been previously reported [14,47], especially that the gut microbiota is involved in the gut–brain axis via neuroendocrine signaling through the hypothalamic–pituitary–adrenal axis, and serotonin regulation [45]. The symptom improvement in the placebo subgroup cannot be explained by the changes in the gut microbiota alone, which means that not only anaerobic bacteria alone but also other factors play an important role in improving IBS symptoms, which is consistent with another publication [46]. 

Our study also has some limitations. Despite the fact that we had a sufficient sample size at the beginning of the study, the patient withdrawals and exclusions after randomization led to a small sample size in each subgroup, which may have caused type II errors in our results. The study included diarrhea-predominant IBS patients but not constipation-predominant, mixed or unspecific types of IBS. The DI was not used in the inclusion criteria for better selection of normobiotic donors and dysbiotic IBS patients.

## 9. Conclusions

In the current study, all transplantation forms were performed safely without any reported complications during the period of both study and post transplantation. Despite the changes observed in different symptom questionnaires, in all three forms of transplantation, the gut microbiota profiles in IBS patients changed after treatment with ACHIM suspension and donor-FMT towards those measured for healthy controls, but not after placebo. Significant changes were found in the bacterial strain signals between placebo vs. ACHIM and FMT. The safety and efficacy of ACHIM suspension and donor-FMT seems similar, but further studies with larger sample size are needed. 

## Figures and Tables

**Figure 1 microorganisms-10-01819-f001:**
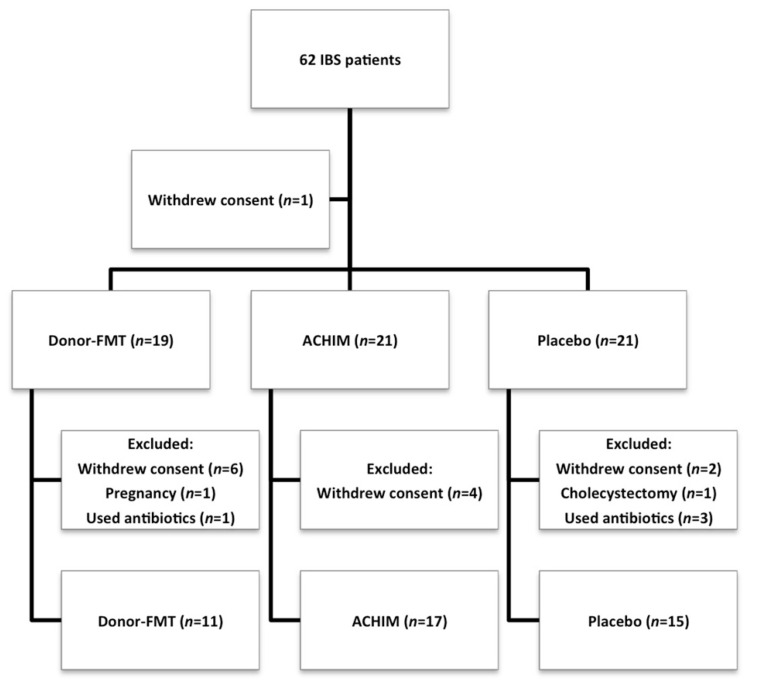
Study flow chart for patients with irritable bowel syndrome (IBS) randomized into three subgroups to receive transplantation with donor fecal microbiota transplantation (donor-FMT), anaerobically cultivated human intestinal microbiota (ACHIM) or placebo.

**Figure 2 microorganisms-10-01819-f002:**
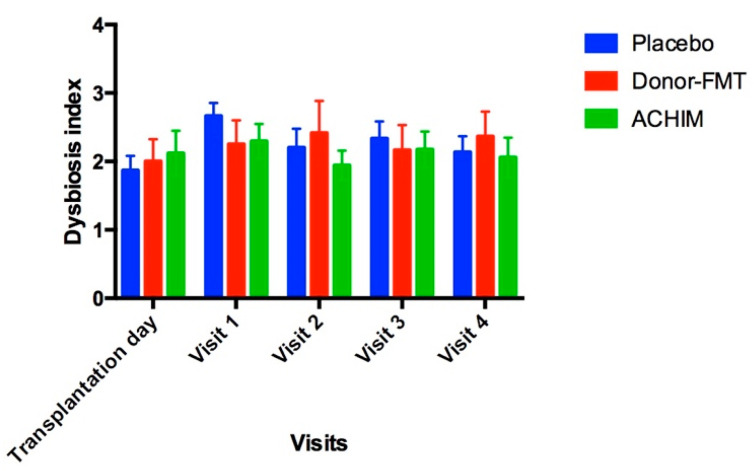
Changes occurring to the Dysbiosis Index before and after receiving transplantation with anaerobically cultivated human intestinal microbiota (ACHIM), donor fecal microbiota transplantation (donor-FMT) or placebo.

**Figure 3 microorganisms-10-01819-f003:**
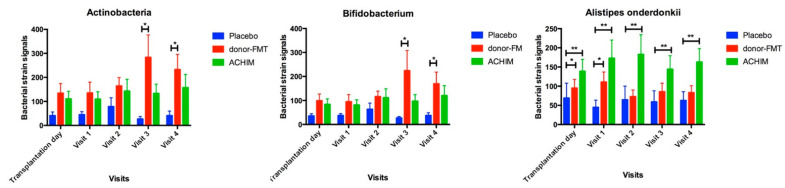
Changes occurring to the bacterial strain signals for *Actinobacteria* spp., *Bifidobacteria* spp. and *Alistipes onderdonkii* before and after receiving transplantation with anaerobically cultivated human intestinal microbiota (ACHIM), donor fecal microbiota transplantation (donor-FMT) or placebo. * *p* < 0.5, ** *p* < 0.01.

**Figure 4 microorganisms-10-01819-f004:**
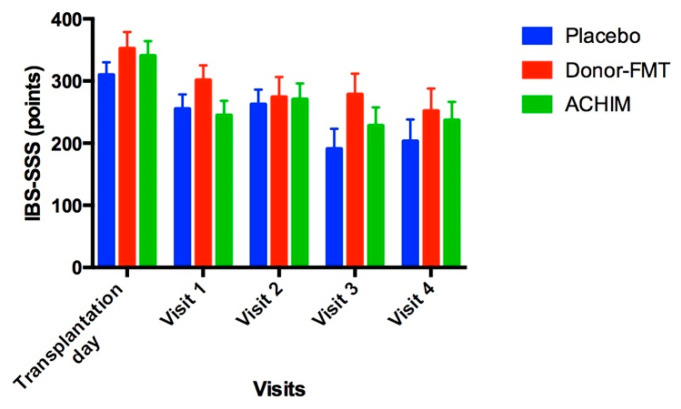
Changes occurring to the IBS-Symptom severity score before and after receiving transplantation with anaerobically cultivated human intestinal microbiota (ACHIM), donor fecal microbiota transplantation (donor-FMT) or placebo.

**Table 1 microorganisms-10-01819-t001:** Changes occurring to the gut microbiota in the total IBS group during the follow up control visits after transplantation with placebo suspension.

Bacteria Strain	Healthy Controls	Transplant Type	Transplantation Day	Visit 1	Visit 2	Visit 3	Visit 4	*p* ^a^	*p* ^b^	*p* ^c^	*p* ^d^	*p* ^e^
*Actinobacteria*	220 ± 71	*Donor-FMT*	134.6 ± 40	135.4 ± 45	164.3 ± 35	283.5 ± 93	232.9 ± 63	>0.9	>0.9	>0.9	>0.9	>0.9
		*ACHIM*	110.6 ± 32	109.4 ± 31	143.2 ± 49	133.4 ± 39	157.4 ± 55	0.8	0.7	>0.9	>0.9	>0.9
		*Placebo*	41 ± 15	45 ± 13	79 ± 37	27 ± 10	41 ± 19	0.057	**0.04**	0.15	**0.01**	**0.03**
*Bifidobacteria*	167 ± 62	*Donor-FMT*	99 ± 28	95 ± 30	116 ± 23	225 ± 84	170 ± 49	>0.9	>0.9	>0.9	>0.9	>0.9
		*ACHIM*	84 ± 23	81 ± 21	112 ± 37	97 ± 27	121 ± 41	0.9	0.6	>0.9	>0.9	0.9
		*Placebo*	35 ± 9	37 ± 8	63 ± 26	27 ± 6	37 ± 11	**0.03**	**0.03**	0.1	**0.009**	**0.04**
*Alistipes onderdonkii*	132 ± 33	*Donor-FMT*	95 ± 23	111 ± 26	73 ± 17	85 ± 22	83 ± 18	>0.9	>0.9	0.9	>0.9	>0.9
		*ACHIM*	139 ± 31	173 ± 47	183 ± 51	144 ± 35	163 ± 35	>0.9	>0.9	>0.9	>0.9	>0.9
		*Placebo*	70 ± 39	45 ± 19	65 ± 36	59 ± 29	63 ± 22	**0.01**	**0.009**	**0.008**	**0.01**	**0.04**

Data are presented as the mean ± SEM. Comparison: Kruskal–Wallis with Dunn’s post-test. **^a^** Healthy controls vs. patients on transplantation day before placebo transplantation, **^b^** Healthy controls vs. patients at 1st visit after placebo transplantation, **^c^** Healthy controls vs. patients at 2nd visit after placebo transplantation, **^d^** Healthy controls vs. patients at 3rd visit after placebo transplantation, **^e^** Healthy controls vs. patients at 4th visit after placebo transplantation. FMT: fecal microbiota transplantation; ACHIM: anaerobically cultivated human intestinal microbiota.

**Table 2 microorganisms-10-01819-t002:** Questionnaire comparison of symptoms before transplantation and after transplantation at each control visit for patients that received anaerobically cultivated human intestinal microbiota.

Questionnaire	Transplantation Day	Visit 1	Visit 2	Visit 3	Visit 4	*p* ^a^	*p* ^b^	*p* ^c^	*p* ^d^
**IBS-SSS**	340.3 ± 24	244.7 ± 23	270.4 ± 26	228 ± 29	236.7 ± 30	**<0.0001**	**0.001**	**0.003**	**0.0008**
**Bristol stool form**	5 ± 0.4	4.7 ± 0.3	4.3 ± 0.4	4.3 ± 0.3	4.4 ± 0.4	0.4	**0.04**	**0.03**	0.057
**SF-NDI**	33.9 ± 1.5	26 ± 2	25.7 ± 2	24.1 ± 2	26.3 ± 2	**0.001**	**0.0003**	**0.002**	**0.008**
**HAD total**	15.2 ± 1.6	15.13 ± 1.4	14.2 ± 1.5	15 ± 1.6	12.6 ± 1.4	0.9	0.4	0.9	0.1
**HAD anxiety**	9.2 ± 1.2	8.8 ± 0.9	7.8 ± 0.9	9.2 ± 1.2	7.2 ± 0.9	0.6	0.1	>0.9	0.06
**HAD depression**	5.8 ± 0.8	6.1 ± 0.8	5.9 ± 0.8	5.6 ± 0.9	5.1 ± 0.8	0.6	0.8	0.9	0.5
**EPQ-N-12**	7.3 ± 0.6	5.2 ± 0.7	6.2 ± 0.8	6.1 ± 0.8	4.9 ± 0.8	**0.005**	**0.04**	0.1	**0.001**

Data are presented as the mean ± SEM. Comparison: Paired *t* test. **^a^** Patients on transplantation day before ACHIM vs. patients at 1st visit after ACHIM, **^b^** Patients on transplantation day before ACHIM vs. patients at 2nd visit after ACHIM, **^c^** Patients on transplantation day before ACHIM vs. patients at 3rd visit after ACHIM, **^d^** Patients on transplantation day before ACHIM vs. patients at 4th visit after ACHIM. FMT: fecal microbiota transplantation; IBS-SSS irritable bowel syndrome-symptom severity score; SF-NDI: short form-Nepean dyspepsia index; HAD: Hospital Anxiety Depression; EPQ-N-12: Eysenck Personality Questionnaire Neuroticism. ACHIM: anaerobically cultivated human intestinal microbiota.

**Table 3 microorganisms-10-01819-t003:** Questionnaire comparison of symptoms before transplantation and after transplantation at each control visit for patients that received fecal microbiota transplantation.

Questionnaire	Transplantation Day	Visit 1	Visit 2	Visit 3	Visit 4	*p* ^a^	*p* ^b^	*p* ^c^	*p* ^d^
**IBS-SSS**	352.1 ± 27	301.4 ± 24	274.3 ± 32	278.6 ± 33	251.9 ± 36	**0.03**	0.2	**0.04**	**0.04**
**Bristol stool form**	5.3 ± 0.3	5.1 ± 0.3	4.5 ± 0.5	4.8 ± 0.3	4.4 ± 0.4	0.6	0.2	0.2	0.04
**SF-NDI**	32.8 ± 2.3	27.8 ± 2.2	27.1 ± 2.5	27.1 ± 3.2	27.2 ± 2.5	**0.004**	**0.02**	0.051	0.052
**HAD total**	12.5 ± 2.3	12 ± 2.3	11.4 ± 2.3	11.89 ± 2.4	11.56 ± 2.9	0.5	0.4	0.4	0.5
**HAD anxiety**	7.3 ± 1.1	7.1 ± 1.1	6.7 ± 1.3	6.9 ± 1.3	7.7 ± 1.7	0.7	0.4	0.2	>0.9
**HAD depression**	5.2 ± 1.6	4.7 ± 1.6	4.7 ± 1.4	4.9 ± 1.4	3.9 ± 1.3	0.2	0.5	0.5	0.3
**EPQ-N-12**	6.8 ± 1.1	5.4 ± 1.1	4.9 ± 1	4.6 ± 1.2	5.2 ± 1.2	**0.02**	0.05	**0.02**	0.1

Data are presented as the mean ± SEM. Comparison: Paired *t* test. **^a^** Patients on transplantation day before FMT vs. patients at 1st visit after FMT, **^b^** Patients on transplantation day before FMT vs. patients at 2nd visit after FMT, **^c^** Patients on transplantation day before FMT vs. patients at 3rd visit after FMT, **^d^** Patients on transplantation day before FMT vs. patients at 4th visit after FMT. FMT: fecal microbiota transplantation; IBS-SSS irritable bowel syndrome-symptom severity score; SF-NDI: short form-Nepean dyspepsia index; HAD: Hospital Anxiety Depression; EPQ-N-12: Eysenck Personality Questionnaire Neuroticism.

**Table 4 microorganisms-10-01819-t004:** Questionnaire comparison of symptoms before transplantation and after transplantation at each control visit for patients that received placebo.

Questionnaire	Transplantation Day	Visit 1	Visit 2	Visit 3	Visit 4	*p* ^a^	*p* ^b^	*p* ^c^	*p* ^d^
**IBS-SSS**	309.8 ± 20	255.1 ± 23	262.6 ± 24	190.9 ± 32	203.1 ± 35	**0.03**	**0.049**	**0.001**	**0.005**
**Bristol stool form**	4.9 ± 0.4	4 ± 0.4	4.313 ± 0.4	3.9 ± 0.4	4 ± 0.4	**0.01**	**0.03**	**0.01**	**0.04**
**SF-NDI**	34.7 ± 1.7	29.6 ± 1.8	27.8 ± 2	23.1 ± 2	22.4 ± 2	**0.02**	**0.002**	**0.005**	**0.007**
**HAD total**	9.2 ± 1.6	8 ± 1.3	7.9 ± 1.6	6.6 ± 1.1	5.9 ± 1.4	0.1	0.053	0.1	0.09
**HAD anxiety**	6.3 ± 1.1	5.3 ± 0.8	5.1 ± 0.9	4.9 ± 0.8	3.9 ± 0.8	0.08	**0.009**	0.2	0.07
**HAD depression**	3.3 ± 0.8	3 ± 0.8	3.1 ± 1	2.1 ± 0.6	2.2 ± 0.9	0.4	0.7	0.1	0.4
**EPQ-N-12**	3.7 ± 0.8	3,5 ± 0.8	3.5 ± 0.9	2.7 ± 9.7	1.7 ± 0.5	0.6	0.5	0.055	**0.006**

Data are presented as the mean ± SEM. Comparison: Paired *t* test. **^a^** Patients on transplantation day before placebo transplantation vs. patients at 1st visit after placebo transplantation, **^b^** Patients on transplantation day before placebo transplantation vs. patients at 2nd visit after placebo transplantation, **^c^** Patients on transplantation day before placebo transplantation vs. patients at 3rd visit after placebo transplantation, **^d^** Patients on transplantation day before placebo transplantation vs. patients at 4th visit after placebo transplantation. FMT: fecal microbiota transplantation; IBS-SSS irritable bowel syndrome-symptom severity score; SF-NDI: short form-Nepean dyspepsia index; HAD: Hospital Anxiety Depression; EPQ-N-12: Eysenck Personality Questionnaire Neuroticism.

## Data Availability

All data are presented in this study.

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
