# Peer review of "The Effect of Anaerobically Cultivated Human Intestinal Microbiota Compared to Fecal Microbiota Transplantation on Gut Microbiota Profile and Symptoms of Irritable Bowel Syndrome, a Double-Blind Placebo-Controlled Study"

_microorganisms, 2022, doi:10.3390/microorganisms10091819_

Round 1

Reviewer 1 Report

Comments to Author:

The manuscript "The effect of anaerobically cultivated human intestinal microbiota compared to fecal microbiota transplantation on gut microbiota profile and symptoms of irritable bowel syndrome, a double-blind placebo-controlled study" of Mazzawi and colleagues provides a double-blind placebo-controlled study on the improvement of FMT in patients with IBS, which is very interesting and provides new evidence for the role of FMT in improving IBS. This is interesting but it reserves many problems:

1. The materials and methods of the manuscript are explained in great detail, but little is written about the specific experimental results, and the discussion is not profound. In terms of effective word count, the Materials and Methods section is much larger than the Results section.

2. The "ACHIM suspension" should be described in detail in Materials and Methods, and the other unimportant ones should be briefly introduced. For example, the species composition of "ACHIM suspension" is specific to the genus level or species level; the order of magnitude of bacteria is 30 x 109 CFU from the literature, so what is the order of magnitude of bacteria in this experiment?

3. Lines 66-67, "ACHIM suspension is a natural ecosystem of normal intestinal flora originating from a healthy donor in 1996, cultivated and propagated regularly in the laboratory (in vitro)". Is it really that reliable after so many years the stability of "ACHIM suspension"? How was it cultivated, what medium was used for cultivation, and has the composition and function not changed since 1996?

4. The highlight of this experiment lies in the efficacy and safety of "ACHIM suspension", donor-FMT, and patient autologous-FMT in the treatment of IBS. First of all, 16S amplicon or metagenome should be used to analyze the structural composition of the three. Further analysis was performed for each FMT course compared to baseline.

5. Why only focus on these three bacteria, Actinobacteria, Bifidobacteria and Alistipes onderdonkii. 16S amplicon or metagenome should be used to analyze diversity from an overall perspective, and then refined to specific strain changes (such as beneficial bacteria, harmful bacteria).

6. This experiment has good plasticity. It is difficult to obtain patient samples and various donors, so the author can redesign the experiment and use metagenomics, proteomics, metabolomics and other multi-omics joint analysis. And finally carry out animal experiments to verify, I believe that exciting discoveries can be obtained in the end.

7. Lines 279-281, “Changes occurring to the IBS-Symptom severity score before and after receiving transplantation with Anaerobically Cultivated Human Intestinal Microbiota (ACHIM), donor fecal microbiota transplantation (donor-FMT) or placebo”. When abbreviations appear for the first time, write their full names, followed by abbreviations.

8. Writing and grammar need to be improved.

Author Response

Dear Editor,

Thank you for the reviewers’ comments which we believe have improved the manuscript. Following is point-by-point response to the reviewers:

  • Reviewer 1:

The manuscript "The effect of anaerobically cultivated human intestinal microbiota compared to fecal microbiota transplantation on gut microbiota profile and symptoms of irritable bowel syndrome, a double-blind placebo-controlled study" of Mazzawi and colleagues provides a double-blind placebo-controlled study on the improvement of FMT in patients with IBS, which is very interesting and provides new evidence for the role of FMT in improving IBS. This is interesting but it reserves many problems:

  1. The materials and methods of the manuscript are explained in great detail, but little is written about the specific experimental results, and the discussion is not profound. In terms of effective word count, the Materials and Methods section is much larger than the Results section.

-Thank you for the comment. The section of material and methods was long due to the complexity of the study involving transplantation of three suspensions. We ought to explain the whole transplantation process for each suspension in addition to patients’ eligibility (inclusion and exclusion criteria), stool collection process and microbial analysis and questionnaires used.  The results section responded to each section of the materials and methods namely; patients included, microbial analysis, questionnaires results, post transplantation complication, with tables and figures too.

  1. The "ACHIM suspension" should be described in detail in Materials and Methods, and the other unimportant ones should be briefly introduced. For example, the species composition of "ACHIM suspension" is specific to the genus level or species level; the order of magnitude of bacteria is 30 x 109CFU from the literature, so what is the order of magnitude of bacteria in this experiment?

-Thank you for the comment. ACHIM has been thoroughly explained under Introduction, lines 64-77. According to the manufacturers ACHIM continues to retain the same biodiversity after 20 years of cultivation therefore the magnitude of bacteria in this experiment should be stable at 30 x 109 CFU.

  1. Lines 66-67, "ACHIM suspension is a natural ecosystem of normal intestinal flora originating from a healthy donor in 1996, cultivated and propagated regularly in the laboratory (in vitro)". Is it really that reliable after so many years the stability of "ACHIM suspension"? How was it cultivated, what medium was used for cultivation, and has the composition and function not changed since 1996?

 -Thank you for your comment. We are not the manufacturers of ACHIM. The manufacturing details of ACHIM are mentioned in reference 23 (A. Gustafsson, A. Berstad, S. Lund-Tonnesen, T. Midtvedt, and E. Norin, “The effect of faecal enema on five microflora-associated characteristics in patients with antibiotic-associated diarrhoea,” Scand J Gastroenterol, vol. 34, no. 6, pp. 580–586, 1999). We preferred not to re-write it to avoid any risk for plagiarism. But briefly, A homogenized fecal suspension was inoculated anaerobically, with resazurin as an indicator, on a 30 ml peptone-yeast medium (Difco Laboratories, USA) containing cholesterol (1.25% freeze-dried hen yolk, Fresenius-Kabi, Sweden), and has since then been re-cultivated every second week and stored at -70°C. Suspensions have been repeatedly tested by molecular methods and all important major groups of microorganisms present in the original fecal sample are still to be found. Additionally, no egg allergens are found to be present. The original feces sample has been kept frozen at -70°C.

The suspension has been used in hundreds of patients with recurrent CDI in Norway and Sweden with a success rate comparable to transplantation using fresh feces (Midtvedt T, Norin E, Benno P, Dahlgren AL. Response to Surawicz et al. Am J Gastroenterol 2013;108:1931-2).

ACHIM suspension has also been previously used for IBS treatment as mentioned in ref. 22 (P. Benno, E. Norin, T. Midtvedt, and P. M. Hellström, “Therapeutic potential of an anaerobic cultured human intestinal microbiota, ACHIM, for treatment of IBS,” Best Pr. Res Clin Gastroenterol, vol. 40–41, p. 101607, 2019, doi: 10.1016/j.bpg.2019.03.003).

  1. The highlight of this experiment lies in the efficacy and safety of "ACHIM suspension", donor-FMT, and patient autologous-FMT in the treatment of IBS. First of all, 16S amplicon or metagenome should be used to analyze the structural composition of the three. Further analysis was performed for each FMT course compared to baseline.

- Thank you for the comment. We used 16s RNA sequencing in this study. Unfortunately, we do not have 16S amplicon or metagenome. But certainly, it would have been a great added value.

  1. Why only focus on these three bacteria, ActinobacteriaBifidobacteriaand Alistipes onderdonkii. 16S amplicon or metagenome should be used to analyze diversity from an overall perspective, and then refined to specific strain changes (such as beneficial bacteria, harmful bacteria).

- These 3 bacteria were the ones that showed most common changes among the three groups during the study. Regarding the use of amplicon, please refer to previous answer.

  1. This experiment has good plasticity. It is difficult to obtain patient samples and various donors, so the author can redesign the experiment and use metagenomics, proteomics, metabolomics and other multi-omics joint analysis. And finally carry out animal experiments to verify, I believe that exciting discoveries can be obtained in the end.

 -Thank you for your comment. Currently we are unable to perform all of these studies due to financial issues. But we will consider them in the next project.

  1. Lines 279-281, “Changes occurring to the IBS-Symptom severity score before and after receiving transplantation with Anaerobically Cultivated Human Intestinal Microbiota (ACHIM), donor fecal microbiota transplantation (donor-FMT) or placebo”. When abbreviations appear for the first time, write their full names, followed by abbreviations.

- Thank you for the comment. The first mention of these names was under introduction, lines 52 and 64. The phrase above is a caption of figure 4, and is written as recommended. 

  1. Writing and grammar need to be improved.

- Thank you for the comment. Improvements have been applied.

Reviewer 2 Report

The manuscript  was a double-blind lacebo-controlled study, and it compared the efficacy and safety of ACHIM suspension with donor-FMT and placebo (patient’s own feces) to treat IBS. There is a conclusion that the safety and efficacy of ACHIM and donor-FMT seems similar in the current study. Therefore, it is very interesting and meaning. Besides, eligible patients and transplantation procedure were strict and appropriate.

Author Response

Dear Editor,

Thank you for the reviewers’ comments which we believe have improved the manuscript. Following is point-by-point response to the reviewers:

The manuscript was a double-blind placebo-controlled study, and it compared the efficacy and safety of ACHIM suspension with donor-FMT and placebo (patient’s own feces) to treat IBS. There is a conclusion that the safety and efficacy of ACHIM and donor-FMT seems similar in the current study. Therefore, it is very interesting and meaning. Besides, eligible patients and transplantation procedure were strict and appropriate.

 -Thank you for the comment.

Reviewer 3 Report

Study on the analysis of the effects of fecal microbiota transplantation (FMT) to improve irritable bowel syndrome (IBS) and changes the profile of the gut microbiota . The comparison involves 3 groups differing in transplant origin : Anaerobically Cultivated Human Intestinal Microbiota (ACHIM),  suspension with donor-FMT and placebo (patient’s own feces).

The manuscript is well written, clear and well presented (minor changes to the figures are suggested in the pdf attached).

Study design is robust, sample size well justified and drawbacks identified and discussed. 

Data presentation in the results and statistical analysis is sound and supports the conclusions.

The subject (ACHIM in IBD patients) is novel and has impact in treatment decisions for these patients. Therefor the manuscript is of interest to the readers.

Minor formatting aspects such as missing italics should be addressed

Author Response

Dear Editor,

Thank you for the reviewers’ comments which we believe have improved the manuscript. Following is point-by-point response to the reviewers:

Study on the analysis of the effects of fecal microbiota transplantation (FMT) to improve irritable bowel syndrome (IBS) and changes the profile of the gut microbiota. The comparison involves 3 groups differing in transplant origin: Anaerobically Cultivated Human Intestinal Microbiota (ACHIM), suspension with donor-FMT and placebo (patient’s own feces).

The manuscript is well written, clear and well presented (minor changes to the figures are suggested in the pdf attached).

- Thank you for the comment.  Changes have been applied as requested.

Study design is robust, sample size well justified and drawbacks identified and discussed. 

- Thank you for the comment.

Data presentation in the results and statistical analysis is sound and supports the conclusions.

- Thank you for the comment.

The subject (ACHIM in IBS patients) is novel and has impact in treatment decisions for these patients. Therefor the manuscript is of interest to the readers.

- Thank you for the comment.

Minor formatting aspects such as missing italics should be addressed.

-Thank you for the comment.

Reviewer 4 Report

Dear sirs,  

Here are my comments:  

The manuscript compares the efficacy and safety of Anaerobically Cultivated Human Intestinal Microbiota and fecal transplantation in 43 IBD patients. The study was performed as randomized, double-blind, and placebo-controlled. IBS symptoms and changes in gut microbiota profile were monitored. Although the number of patients was quite modest certain positive changes were observed. the manuscript is well written and  designed. Some modifications should however performed  

comments  

1. please change the title to be shorter

2. line 14. please change the word improves tp may improve or has been shown to improve. There is no enough evidence to claim so strongly

3a.  please check the writing of bacterial taxa. e.g. change Actinobacteria to actinobacteria or use italics and spp. annex

3b.  please check one bacterium or many bacteria i.e. bifidobacteria or Bifidobacteria spp. in italics.

4.  lines 25/ 27/31. please specify the word " signal / signals "

5. line 63. please give more specific time frames than several months. it is highly important how long the improvements last and why not longer should be discussed at least in discussion. 

Author Response

Dear Editor,

Thank you for the reviewers’ comments which we believe have improved the manuscript. Following is point-by-point response to the reviewers:

The manuscript compares the efficacy and safety of Anaerobically Cultivated Human Intestinal Microbiota and fecal transplantation in 43 IBD patients. The study was performed as randomized, double-blind, and placebo-controlled. IBS symptoms and changes in gut microbiota profile were monitored. Although the number of patients was quite modest certain positive changes were observed. The manuscript is well written and designed. Some modifications should however be performed  

comments  

  1. please change the title to be shorter

- Thank you for the comment. The title is consistent with the guidelines of the journal and it is important to reflect the content of study in the title.

  1. line 14. please change the word improves to may improve or has been shown to improve. There is no enough evidence to claim so strongly

- Thank you for the comment. Changes have been applied as requested.

3a.  please check the writing of bacterial taxa. e.g. change Actinobacteria to actinobacteria or use italics and spp. annex

-Thank you for the comment. Changes have been applied as requested.

3b.  please check one bacterium or many bacteria i.e. bifidobacteria or Bifidobacteria spp. in italics.

-Thank you for the comment. Changes have been applied as requested.

  1.  lines 25/ 27/31. please specify the word " signal / signals "

-Thank you for the comment. Changes have been applied as requested.

  1. line 63. please give more specific time frames than several months. it is highly important how long the improvements last and why not longer should be discussed at least in discussion. 

-Thank you for the comment. Changes have been applied as requested. The maximum reported FMT effect was reported by another group up to one year due to the study design. No data to tell us if the effect would last longer than that yet. In our study, it was seven months due to the design of our study.

Round 2

Reviewer 1 Report

The author sincerely and seriously replied to the reviewer's question and answered the reviewer's confusion.

It is hoped that the preservation and efficacy of "ACHIM" are as described by the author, and no major problems have been found at present. The manuscript can be published after the author's own minor repairs and improvements for the language and format of the manuscript.